# Porcupettes Management at Wildlife Rescue Centers and Liberation into the Wild: Implications for Post-Liberation Success

**DOI:** 10.3390/ani13091546

**Published:** 2023-05-05

**Authors:** Francesca Coppola, Chiara Dari, Giuseppe Vecchio, Marco Aloisi, Giorgia Romeo, Claudia Biliotti, Antonio Felicioli

**Affiliations:** 1Department of Veterinary Sciences, University of Pisa, Viale delle Piagge 2, 56124 Pisa, Italy; francesca.coppola@vet.unipi.it (F.C.); chiara_dari@yahoo.it (C.D.); 2Studio Agrofauna, Via delle Corellaie 1, 57121 Livorno, Italy; vecchio@agrofauna.it; 3CRASM Semproniano, Loc. Casaccia snc, 58055 Semproniano, Italy; marcoaloisi@yahoo.it (M.A.); biliotti.cla@gmail.com (C.B.); 4Wildlife Sector, Tuscany Regional Council, Via Trieste 5, 58100 Grosseto, Italy; giorgina.romeo@regione.toscana.it

**Keywords:** wildlife rescue center, *Hystrix cristata*, management strategies, liberation protocol, post-liberation monitoring

## Abstract

**Simple Summary:**

Wildlife rescue centers play a key role for crested porcupine rescue and conservation. There is a lack of guidelines for management and liberation, which is a crucial issue, especially for porcupettes to ensure a suitable behavioral development for wildlife. In this work, preliminary data on the post-liberation behavior and survival of rescued porcupines under different management and liberation strategies were collected, providing the first useful knowledge base and fulfilling the urgency to develop official guidelines.

**Abstract:**

The crested porcupine is a frequent host species in wildlife rescue centers and no guidelines for its management, liberation and post-liberation monitoring are yet available. Here, captive-grown porcupines’ behavior and survival in the wild after liberation were investigated and described for the first time. Management strategies adopted at the centers could affect porcupine adaptation to the natural environment. The detention of porcupettes in single cages and fed only human-supplied food may not ensure ethological welfare, nor a suitable behavioral development compatible with wildlife. The liberation of captive-grown porcupines should also be carefully planned to promote and increase the possibility of post-liberation success.

## 1. Introduction

Wildlife Rescue Centers (WRC) play a key role for wildlife rescue and conservation. Their mission is the rehabilitation and the release back into the wild of injured or orphaned hospitalized wild animals able to survive and reproduce [1,2]. The crested porcupine (*Hystrix cristata* L., 1758) is a widely distributed burrowing Italian rodent [3,4,5,6] strictly protected by both the European (Bern Convention, 1979) and Italian law (N. L. 503/1981). The number of porcupines hospitalized in WRC every year range from 10 to 25, mainly involving adults or subadults but also porcupettes (Ceccarelli R. and Aloisi M., Personal communication). The hospitalization period length usually ranges from 5 to 60 days for adult individuals and up to 1 year for porcupettes, with mortality rates reaching up to 90% due to the severity of injuries and a very low release and/or liberation rate (10–15%). Different translocation tactical options have been outlined to support the IUCN/SSC Guidelines for Reintroduction and other Conservation Translocations which represent a valuable resource that remains open to improvement from new field data [7].

For porcupines, no management guidelines at WRC or liberation protocols are available yet. As for other mammalian species, a rehabilitation or pre-adaptation phase to the natural environment is usually not planned and any kind of post-liberation monitoring is performed. 

In this pilot study, the post-liberation behavior and survival in the wild of two captive-grown porcupines under different WRC management and liberation protocols were preliminarily investigated and described. In this paper, authors refer to the term “release” as giving “freedom” to animals, which before hospitalization were already free (i.e., adult and/or sub-adult individuals). The authors refer to the term “liberation” as giving “freedom” to animals born and/or grown in captivity (i.e., porcupettes between 0 and 3 months old).

## 2. Materials and Methods

### 2.1. Experimental Animals

Post-liberation data collection was performed on two captive-grown porcupines in rescue centers: a male (M1) and a female (F2). M1 was hospitalized in the center at the age of about 3 months old. It was always detained in a wire mesh cage (150 × 250 × 170 cm) supplied with a dog kennel with straw as a refuge site (Figure 1A) and fed only with corn, flakes cereal, vegetables, and fruit.

F2 was extracted by the dead mother after an impact with a car and hospitalized in the center just after being born with the umbilical cord still attached. During the first three months of hospitalization, it was kept in a cage and fed powdered milk (Esbilac^®^) every three hours. At about three months, it was weaned with vegetables and fruits and then transferred into a natural open-air enclosure (about 200 m^2^) with only land, grass, and a refuge site of reeds (Figure 1B) where it lived until it was liberated with 7 other co-specifics. Corn, vegetables, and fruit were provided only as supplementary food to the natural ones present within the enclosure.

Both M1 and F2 were liberated at about 14 months and had a weight of 8.6 kg and 9.1 kg, respectively.

### 2.2. Liberation Protocols

Porcupine liberation was performed in a hilly area of 500 ha (43°35′4.12″ N–10°32′50.52″ E) in the municipality of Crespina-Lorenzana in Pisa Province, Tuscany, Italy (Figure 2). The area is characterized by an anthropic, fragmented agroecosystem where small woody areas intersperse with cultivated and/or uncultivated areas. In the area, the crested porcupine has been present with a stable population since historical times and 26 settlements are known. All known settlements were monitored by camera traps for 1 month before each liberation event. The liberation settlements were chosen among those not inhabited by resident porcupines and/or other mammals. 

Both porcupines were liberated directly in the burrow by placing the wooden case with the animal inside in front of a ground entrance hole. 

Before liberation, each porcupine was weighed, anaesthetized [8], sexed, individually marked with colored adhesive tape applied on quills, and equipped with a radio-collar (Figure 3). M1 was liberated on 9 September 2012 at 7:00 p.m. while F2 was on 8 October 2015 at 12:30 a.m. Potatoes and courgettes were provided in the surroundings of the F2 liberation site for the first month of post-liberation.

### 2.3. Post-Liberation Monitoring and Data Analysis

The post-liberation monitoring of porcupine behavior and survival was performed by using satellite and radio telemetry and camera trapping. GPS radio-collars with GSM technology (Lotek, WildCell SLG) were supplemented with a VHF radio-collar (Biotrack TW3, Wareham, UK) with a total weight of 260 g. The collar was set up to record 13 localizations (fixes) per day, 7 during the night and 5 during daylight hours, and to send a message after every 5–6 fixes were recorded. The mortality sensor of the collar was also activated.

Radio telemetry was used as an integrative tool for each porcupine’s localization whenever no data for more than one day were obtained from the GPS collar. A homing-in technique was used for animal localization, and for each localization, the geographic coordinates were recorded by the GPS receiver (GPS Garmin Oregon 550^®^). Fixes with geographical localization (FGL) obtained for M1 and F2 were used for analyzing porcupines’ usage of space (i.e., home range, daylight, and nocturnal motor activity) using open source Q-Gis 2.18 software. The porcupines’ home range was calculated using the adaptive kernel to 95% [9,10]. Fixes recorded from 7:00 a.m. to 4:00 p.m. were selected for daylight motor activity analysis, while fixes recorded from 5:00 p.m. to 6:00 a.m. were selected for nocturnal activity. The number of diurnal and nocturnal fixes collected in different habitat types (i.e., woody or ecotone strips, uncultivated areas, vineyards, and olive groves) were analyzed using the chi-square test (χ^2^).

For camera-trapping activity, eighteen camera traps with passive infrared sensors were used. The camera-trapping activity was continuously performed either in each liberation site, in another two settlements near that of liberation, and in eight sites randomly chosen along pathways and in food-patch areas near monitored settlements. The camera traps were set to record 20 s video clips without time-lapse. Videos recorded by camera traps were checked and analyzed daily. Burrowing, feeding, digging and reproductive behaviors were used as the criteria to disclose the liberation success.

## 3. Results

Overall, 16.011 camera trap videos were collected and analyzed, of which there were 7.135 where animals were present. Video recordings allowed for us to assess that M1 left the burrow just two hours after liberation and did not come back there anymore (Figure 4A). In the first week post-liberation, M1 was caught on camera traps near the settlement of liberation on only two occasions. Subsequently, it left the liberation site and never returned. Ten days post-liberation, it was found under a strongly debilitated bush (i.e., lethargic and without showing any defense and escape behavior to human presence) and with a quill embedded in one cheek. It was then recaptured, fed for two days, and then re-liberated in the same settlement on the 22 September at 4:00 p.m. After the second liberation, M1 remained in the burrow all day, emerged at 9:00 p.m., and never came back again. From this moment onward, it performed a wandering life within the study area, never using a burrow as a daytime refuge; it was not possible to collect data on its feeding, digging and reproductive behavior. On the 18 November 2012 at 3:00 a.m., 70 days from liberation, the GPS collar sent the mortality message. In the mortality site, a lot of M1-marked quills and a radio-collar visibly damaged by porcupine teeth above a roe deer horn were detected, while the animal body was never found.

Conversely, after liberation, F2 remained for eight consecutive days in the burrow without ever emerging. On 15 October 2015 at 10:40 p.m., it emerged from the burrow for the first time and came back to the same burrow on 16 October 2015 at 3:09 a.m. (Figure 4B). For the first 17 days of post-liberation, it always remained in the woody area of liberation and fed on leftover food available in the surroundings of the settlement but also on grass and roots (Figure 5A). Subsequently, it began to visit pasture areas and only occasionally fed on artificially supplied food. After 20 days of liberation, F2 left the liberation site and moved within the study area, using bramble bushes as a daytime refuge. After 23 days of wandering life, it returned to the liberation settlement with a male specimen with which it established a stable pair relationship, showing all the reproductive behavioral displays described by Felicioli et al. [11,12] and Coppola and Felicioli [13] (Figure 5B). From this moment onward, F2 and the male always lived together, inhabiting two monitored settlements alternately. From the camera-trap videos, the digging behavior of F2 was also documented. After 160 days of liberation, F2 was found dead at the exit of the burrow where it lived. Pulmonary edema was identified as the cause of death in a necropsy examination.

From the GPS collar and radio-tracking activity, a total of 466 and 151 FGL were obtained for M1 and F2, respectively. The kernel containing 95% of total home range resulted in 112 ha for M1 and 39 ha for F2. The 16.7% (*n* = 78) of FGL for M1 and the 24.5% (*n* = 37) for F2 were recorded during daylight hours. The daylight fixes of the two porcupines were always recorded in woody areas and/or ecotone strips. For F2, daylight fixes were recorded only during the second month post-liberation, while for M1, this occurred during the entire period of monitoring. For both porcupines, nocturnal activity was significantly higher (*p* < 0.001) in uncultivated areas or ecotone strips (*n* = 297 for M1 and *n* = 107 for F2) than in vineyards (*n* = 8 for M1 and *n* = 3 for F2) and olive groves (*n* = 33 for M1 and *n* = 4 for F2), while it was never recorded in vegetable gardens or cultivated fields.

## 4. Discussion

The data obtained in this study are the first preliminary sets of information on post-liberation behavior and survival in the wild of two captive-grown porcupines. Although the results obtained are not conclusive, they could be a base towards drawing up a first attempt of guidelines for porcupine management at WRC and porcupine liberation.

The differences recorded between the two liberated porcupines in their adaptation ability to natural environments in the use of burrowing, feeding, digging, and reproduction could be affected by different detention conditions at WRC. Moreover, different responses to captive conditions in relation to animal sex could be hypothesized and further investigations on this are desirable. Wildlife detection in unsuitable structures at WRC and the absence of a rehabilitation phase before liberation have been previously reported as the main limiting factors to rehabilitate animals in suitable physical and behavioral conditions [1,14,15]. The crested porcupine is a gregarious species that lives in family groups in which youngsters remain for at least 1 year [13,16,17]. The reasons for the long permanence of sub-adults within the family are unknown. However, a possible hypothesis is that it may be linked to a behavioral learning process. Therefore, a lack of familiar social interaction among porcupettes due to long detention periods in single cages could contribute to unsuccessful post-liberation. Therefore, the detention of porcupettes in natural enclosures with conspecific animals could be a better approach to ensuring suitable animal behavioral development for wildlife and ethological welfare. The porcupine diet is mainly based on roots, bulbs, and tubers [18,19,20,21]. The debilitation state in which M1 was recovered allows for us to hypothesize that an inability to search for food after liberation is a possible cause. This could suggest that long detention periods can lead porcupines to develop addiction towards human-supplied feed, as also observed in wolves and birds [22,23,24,25,26,27,28]. Therefore, M1-post-liberation behavior could lead to other attributes as the causes of unsuccessful liberation. The inability for adaptation to wild environments causing death could be due to the following: (I) stress and debilitation post-liberation; (II) lethal fighting with a wild porcupine, and/or other species; (III) poaching event.

Conversely, the presence of land, plants, and grass in the F2 enclosure may have had a positive effect on its ability to autonomously search for food after liberation. Therefore, the death of F2 is not attributable to difficulties in adaptation to wildlife and is not considered a real post-liberation failure. Pulmonary edema was identified as the cause of death for F2 and was also detected in an additional four wild-found dead porcupines in the study area during the investigation (unpublished data). This evidence raises the urgency to perform further investigations on the impact of this symptomatology on porcupine population mortality. 

The wider home-range size of M1 compared to F2 can be conceived to be the result of the partially fixed collection obtained for F2 compared to M1 due to the different percentages of GPS collar activity times (30.6%, 49/160 vs. 100%, 70/70, respectively) during the survival time of each animal. GPS technology is a highly efficient monitoring technique for many animal taxa [29,30]. However, the use of this technology on crested porcupines results in poor efficiency due to the behavioral and ecological habits of this species (i.e., living in underground burrows and mainly moving in pathways under woody areas with a dense vegetation cover). This determines frequent failure in satellite connection, a high number of collection attempts due to fixes, and with consequential rapid exhaustion of collar batteries, as recorded for F2. 

The liberation protocol adopted may affect the outcome of liberation events. According to Mariacher [1], Botteghi and Fraticelli [31], and Fraissinet [32], animal liberation should be performed in the hours in which the species is active. However, for captive-grown porcupines, liberation during physiological motor inactivity hours [33,34] may be a better tactic to increase animal permanence in the burrow and to promote a less stressed familiarization with the new environment. Moreover, the marked preference for human-supplied food shown by F2 suggests that providing familiar feeding as in WRC may contribute to reducing post-liberation stress and promoting the adaptation process as also reported for raptors [35]. 

## 5. Conclusions

Authors are aware of the small sample size as well as the preliminary and non-exhaustive features of the results obtained. However, these results are a useful new scientific knowledge base for further investigation aimed at developing official management guidelines and a liberation protocol. Both management strategies adopted at the centers and liberation methods could affect porcupine adaptation ability to natural environments. The detention of captive-grown porcupines at WRC should be performed in structures that allow for us to ensure animal ethological welfare and promotion of a suitable behavioral development for wildlife. To this aim, the use of a single wire mesh cage must be avoided, while the adoption of collective natural open-air enclosures with only land, spontaneous vegetation, and natural refuge sites is encouraged. The use of artificial food should be limited and only provided as supplementation to the natural one. Porcupine liberation must be performed during daylight hours in an uninhabited settlement, providing familiar food in the area surrounding the liberation site. 

## Figures and Tables

**Figure 1 animals-13-01546-f001:**
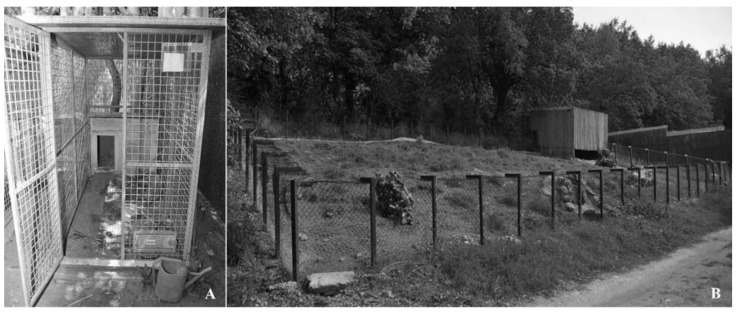
(**A**) The wire mesh cage with the dog kennel and (**B**) the open-air enclosure with the refuge site of reeds in which M1 and F2, respectively, lived at rescue centers.

**Figure 2 animals-13-01546-f002:**
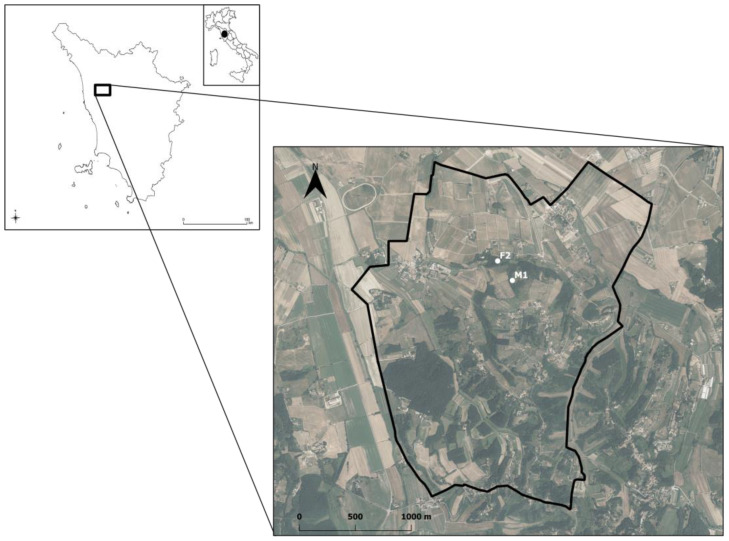
Map of the location of the study area in the Tuscany Region. In detail, the liberation sites of M1 and F2 are shown, respectively.

**Figure 3 animals-13-01546-f003:**
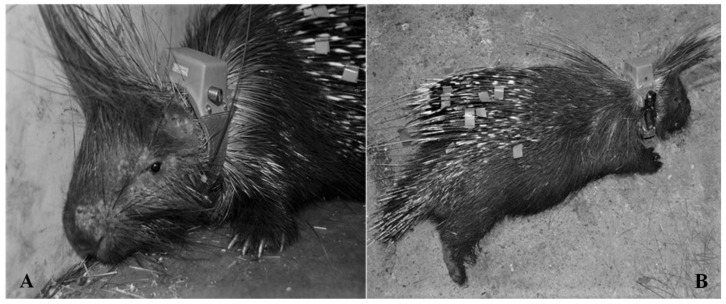
M1 (**A**) and F2 (**B**) were equipped with GPS radio-collars integrated with VHF collars before liberation. Quills marked with colored adhesive tape are also visible.

**Figure 4 animals-13-01546-f004:**
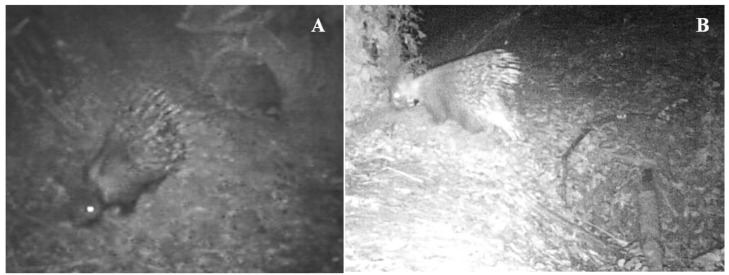
First emergence from the burrow after liberation of M1 (**A**) and F2 (**B**).

**Figure 5 animals-13-01546-f005:**
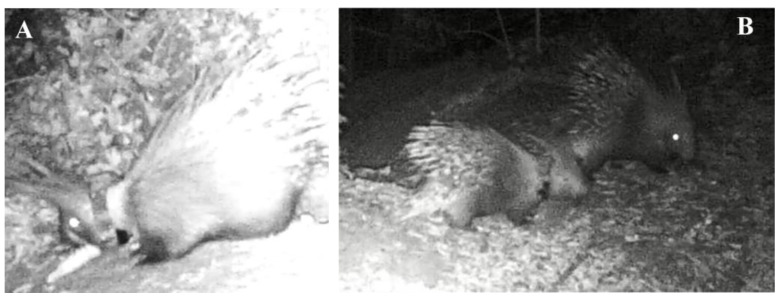
(**A**) F2 while eating a courgette left in the site of liberation and (**B**) while transiting in the liberation settlement with the male with which it established a pair after liberation.

## Data Availability

All data are available upon request to the corresponding author.

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
