# Peer review of "Porcupettes Management at Wildlife Rescue Centers and Liberation into the Wild: Implications for Post-Liberation Success"

_animals, 2023, doi:10.3390/ani13091546_

Round 1

Reviewer 1 Report

This article reported two unsuccessful cases of porcupettes liberation into the wild.  However, this article was over-interpreted based only on too few data. 

Reviewer 2 Report

This study stresses the importance of management, liberation and post-liberation monitoring for rescued porcupines to improve and test their adaptation ability to natural environment. The limit in sample size makes the robustness of the data an issue unless proven otherwise.

1. A thorough review of known conservation reintroduction or translocation should be provided in the Introduction.

2. A map of the study area presenting main habitats and release sites would be welcome. This might be important to explain the behavioral differences between the two porcupines, especially if the release sites were in or surrounded by distinct habitats.

3. Fixes with geographical localization were used for space and habitat use analyses. However, even the authors do not support the effectiveness of GPS technology in this study. Actually, the technological limits would lead to potential bias in the results of home-range and habitat use.

4. Without clear behavior definitions, the behavioral descriptions in Results are confusing, such as “strongly debilitated”, “F2 appeared very excited”, etc.

5. Gender can also influence the differences in many aspects between the two porcupines. To identify this impact, more studied individuals and/or references should be added for a comprehensive comparison.

Reviewer 3 Report

The authors presented a preliminary study to describe different management strategies and liberation protocols on the post-liberation behaviour and survival in porcupines.

I believe this preliminary data is important to be shared with the scientific community, namely vets and professionals from other wildlife rescue centres, even if your N is not considerable. The manuscript is well-written and the conclusions are well-discussed and sustained. However, I believe authors should clearly consider turning this manuscript into a Short communication or Case description, because, in my point of view, it is merely a description of two cases (which, as a said before, deserves its merit) and does not fit into an Original Article structure. Actually, the authors call this as "preliminary". The Results section is the most obvious proof that this change should be addressed, it is very textual. 

Other aspects I have detected:

L42: "To this aim liberation should be performed during locomotory inactivity hours of the species near uninhabited settlements leaving animals to autonomously enter in a burrow and supplying it familiar food." - there are no commas or full stops in this sentence and it makes it very hard to read.

L58: "the two main causes of the species occurrence in WRC" - Authors should provide a reference to this.

L94-109: There is no need to say "porcupine male" and "porcupine female" when the authors already presented these animals as M1 and F2. Please use only these designations (F2 and M1) after introducing them in this section.

Round 2

Reviewer 1 Report

This revised version is much better.

Author Response

Thank you for your comment. Following the Academic Editor suggestion the manuscript has been revised by a native English speaker.

Reviewer 2 Report

Overall, this version is coherent, and authors have polished up all aspects required in initial comments.

Author Response

(The authors gave the same response as above.)

Reviewer 3 Report

The authors have answered and addressed all my comments. Therefore. I have nothing else to add and wish the authors all the best.

Author Response

Thank you for your comment. Following the Academic Editor suggestion the manuscript has been also revised by a native English speaker.